

# A Bayesian brain model of adaptive behavior: an application to the Wisconsin Card Sorting Task

Marco D'Alessandro[1], Stefan T. Radev[2], Andreas Voss[2] and Luigi Lombardi[1]

[1] Department of Psychology and Cognitive Science, University of Trento, Rovereto, Italy
[2] Institute of Psychology, Heidelberg University, Heidelberg, Germany

## ABSTRACT

Adaptive behavior emerges through a dynamic interaction between cognitive agents and changing environmental demands. The investigation of information processing underlying adaptive behavior relies on controlled experimental settings in which individuals are asked to accomplish demanding tasks whereby a hidden regularity or an abstract rule has to be learned dynamically. Although performance in such tasks is considered as a proxy for measuring high-level cognitive processes, the standard approach consists in summarizing observed response patterns by simple heuristic scoring measures. With this work, we propose and validate a new computational Bayesian model accounting for individual performance in the Wisconsin Card Sorting Test (WCST), a renowned clinical tool to measure set-shifting and deficient inhibitory processes on the basis of environmental feedback. We formalize the interaction between the task's structure, the received feedback, and the agent's behavior by building a model of the information processing mechanisms used to infer the hidden rules of the task environment. Furthermore, we embed the new model within the mathematical framework of the Bayesian Brain Theory (BBT), according to which beliefs about hidden environmental states are dynamically updated following the logic of Bayesian inference. Our computational model maps distinct cognitive processes into separable, neurobiologically plausible, information-theoretic constructs underlying observed response patterns. We assess model identification and expressiveness in accounting for meaningful human performance through extensive simulation studies. We then validate the model on real behavioral data in order to highlight the utility of the proposed model in recovering cognitive dynamics at an individual level. We highlight the potentials of our model in decomposing adaptive behavior in the WCST into several information-theoretic metrics revealing the trial-by-trial unfolding of information processing by focusing on two exemplary individuals whose behavior is examined in depth. Finally, we focus on the theoretical implications of our computational model by discussing the mapping between BBT constructs and functional neuroanatomical correlates of task performance. We further discuss the empirical benefit of recovering the assumed dynamics of information processing for both clinical and research practices, such as neurological assessment and model-based neuroscience.

Corresponding author
Marco D'Alessandro,
marco.dalessandro@unitn.it

# INTRODUCTION

Computational models of cognition provide a way to formally describe and empirically account for mechanistic, process-based theories of adaptive cognitive functioning (*Sun, 2009*; *Cooper et al., 1996*; *Lee & Wagenmakers, 2014*). A foundational theoretical framework for describing functional characteristics of neurocognitive systems has recently emerged under the hood of Bayesian brain theories (*Knill & Pouget, 2004*; *Friston, 2010*). Bayesian brain theories owe their name to their core assumption that neural computations resemble the principles of Bayesian statistical inference.

In a Bayesian theoretical framework, cognitive agents interact with an uncertain and changeable sensory environment. This requires a cognitive system to infer sensory contingencies based on an internal generative model of the environment. Such a generative model represents subjective hypotheses, or beliefs, about the causal structure of events in the environment (*Friston, 2005*; *Knill & Pouget, 2004*) and forms a basis for adaptive behavior. It is assumed that internal beliefs are constantly updated and refined to match the current state of the world as new observations become available. The core idea behind the Bayesian brain hypothesis is that computational mechanisms underlying such an internal belief updating follow the logic of Bayesian probability theory. In this respect, information about the external world provided by sensory inputs is represented as a conditional probability distribution over a set of environmental states. Consequently, the brain relies on this probabilistic representation of the world to infer the most likely environmental causes (states) which generate those inputs, and such a process follows the computational principles of Bayesian inference (*Friston & Kiebel, 2009*; *Friston, 2010*; *Buckley et al., 2017*).

To clarify this concept, consider a simple example of a perceptual task in which a cognitive agent is required to judge whether an item depicted on a flat plane is concave or convex. Its judgment is based solely on the basis of a set of observed perceptual features, such as, shape, orientation, texture and brightness. Here, the concave-to-convex gradient entails the set of environmental states which must be inferred. The internal generative model of the agent codifies beliefs about how different degrees of convexity might give rise to certain configurations of perceptual inputs. From a Bayesian perspective, the problem is solved by *inverting* the generative model of the environment in order to turn assumptions about how environmental states generate sensory inputs into beliefs about the most likely states (e.g., degree of convexity) given the available sensory information.

Potentially, there are no limitations regarding the complexity of environmental settings (e.g., items and rules in experimental tasks) and cognitive processes to be described in light of the Bayesian brain framework. Indeed, the latter has proven to be a consistent computational modeling paradigm for the investigation of a variety of neurocognitive mechanisms, such as motor control (*Friston et al., 2010*), oculomotor dynamics (*Friston et al., 2012*), object recognition (*Kersten, Mamassian & Yuille, 2004*), attention (*Feldman & Friston, 2010*), perceptual inference (*Petzschner, Glasauer & Stephan, 2015*; *Knill & Pouget, 2004*), multisensory integration (*Körding et al., 2007*), as well as for providing a foundational theoretical account of general neural systems' functioning (*Lee & Mumford, 2003*; *Friston, 2005*; *Friston, 2003*) and complex clinical scenarios such as Schizophrenia
(*Stephan, Baldeweg & Friston, 2006*), and Autistic Spectrum Disorder (*Haker, Schneebeli & Stephan, 2016*; *Lawson, Rees & Friston, 2014*). For this reason, such a modeling approach might provide a comprehensive and unified framework under which several cognitive impairments can be measured and understood in the light of a general process-based theory of neural functioning.

In this work, we address the challenging problem of modeling adaptive behavior in a dynamic environment. The empirical assessment of adaptive functioning often relies on dynamic reinforcement learning scenarios which require participants to adapt their behavior during the unfolding of a (possibly) demanding task. Typically, these tasks are designed with the aim to figure out how adaptive behavior unfolds through multiple trials as participants observe certain environmental contingencies, take actions, and receive feedback based on their actions. From a Bayesian theoretical perspective, optimal performance in such adaptive experimental paradigms require that agents infer the probabilistic model underlying the hidden environmental states. Since these models usually change as the task progresses, agents, in turn, need to adapt their inferred model, in order to take optimal actions.

Here, we propose and validate a computational Bayesian model which accounts for the dynamic behavior of cognitive agents in the Wisconsin Card Sorting Test (WCST; *Berg, 1948*; *Heaton, 1981*), which is perhaps the most widely adopted neuropsychological setting employed to investigate adaptive functioning. Due to its structure, the WCST can account for executive components underlying observed behavior, such as set-shifting, cognitive flexibility and impulsive response modulation (*Bishara et al., 2010*; *Alvarez & Emory, 2006*). For this reason, we consider the WCST as a fundamental paradigm for investigating adaptive behavior from a Bayesian perspective.

The environment of the WCST consists of a target and a set of stimulus cards with geometric figures which vary according to three perceptual features. The WCST requires participants to infer the correct classification rule by trial and error using the examiner's feedback. The feedback is thought to carry a positive or negative information signaling the agent whether the immediate action was appropriate or not. Modeling adaptive behavior in the WCST from a Bayesian perspective is straightforward, since observable actions emerge from the interaction between the internal probabilistic model of the agent and a set of discrete environmental states.

Performance in WCST is usually measured via a rough summary metric such as the number of correct/incorrect responses or pre-defined psychological scoring criteria (see for instance *Heaton, 1981*). These metrics are then used to infer the underlying cognitive processes involved in the task. A major shortcoming of this approach is that it simply assumes the cognitive processes to be inferred without specifying an explicit *process model*. Moreover, summary measures do not utilize the full information present in the data, such as trial-by-trial fluctuations or various interesting agent-environment interactions. For this reason, crude scoring measures are often insufficient to disentangle the dynamics of the relevant cognitive (sub)processes involved in solving the task. Consequently, an entanglement between processes at the metric level can prevent us from answering interesting research questions about aspects of adaptive behavior.

In our view, a sound computational account for adaptive behavior in the WCST needs to provide at least a quantitative measure of effective belief updating about the environmental states at each trial. This measure should be complemented by a measure of how feedback-related information influences behavior. The first measure should account for the integration of meaningful information. In other words, it should describe how prior beliefs about the current environmental state change after an observation has been made. The second measure should account for signaling the (im)probability of observing a certain environmental configuration (e.g., an (un)expected feedback given a response) (*Schwartenbeck, FitzGerald & Dolan, 2016*).

Indeed, recent studies suggest that the meaningful information content and the pure unexpectedness of an observation are processed differently at the neural level. Moreover, such disentanglement appears to be of crucial importance to the understanding of how new information influences adaptive behavior (*Nour et al., 2018*; *Schwartenbeck, FitzGerald & Dolan, 2016*; *O'Reilly et al., 2013*). Inspired by these results and previous computational proposals (*Koechlin & Summerfield, 2007*), we integrate these different information processing aspects into the current model from an information-theoretic perspective.

Our computational cognitive model draws heavily on the mathematical frameworks of Bayesian probability theory and information theory (*Sayood, 2018*). First, it provides a parsimonious description of observed data in the WCST via two neurocognitively meaningful parameters, namely, *flexibility* and *information loss* (to be motivated and explained in the next section). Moreover, it captures the main response patterns obtainable in the WCST via different parameter configurations. Second, we formulate a functional connection between cognitive parameters and underlying information processing mechanisms related to belief updating and prediction formation. We formalize and distinguish between *Bayesian surprise* and *Shannon surprise* as the main mechanisms for adaptive belief updating. Moreover, we introduce a third quantity, which we named predictive *Entropy* and which quantifies an agent's subjective uncertainty about the current internal model. Finally, we propose to measure these quantities on a trial-by-trial basis and use them as a proxy for formally representing the dynamic interplay between agents and environments.

The rest of the paper is organized as follows. First, the WCST is described in more detail and a mathematical representation of the new Bayesian computational model is provided. Afterwards, we explore the model's characteristics through simulations and perform parameter recovery on simulated data using a powerful Bayesian deep neural network method (*Radev et al., 2020*). We then apply the model to real behavioral data from an already published dataset. Finally, we discuss the results as well as the main strengths and limitations of the proposed model.

## THE WISCONSIN CARD SORTING TEST

In a typical WCST (*Heaton, 1981*; *Berg, 1948*), participants learn to pay attention and respond to relevant stimulus features, while ignoring irrelevant ones, as a function of

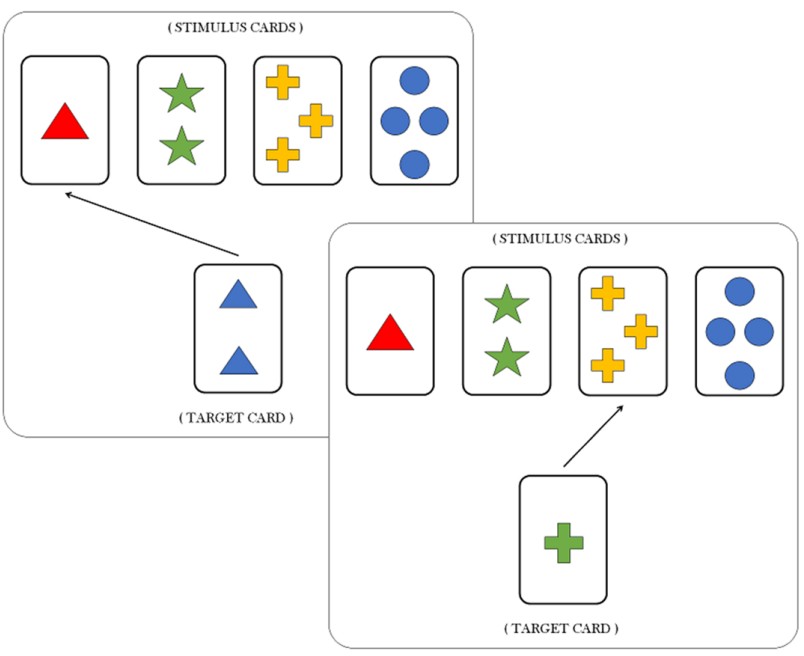

**Figure 1** Suppose that the current sorting rule is the feature shape. The target card in the first trial (left box) contains two blue triangles. A correct response requires that the agent matches the target card with the stimulus card containing the single triangle (arrow represents the correct choice), regardless of the features color and number. The same applies for the second trial (right box) in which matching the target card with the stimulus card containing three yellow crosses is the correct response.

experimental feedback. In particular, Individuals are asked to match a target card with one of four stimulus cards according to a proper sorting principle, or sorting rule. Each card depicts geometric figures that vary in terms of three features, namely, color (red, green, blue, yellow), shape (triangle, star, cross, circle) and number of objects (1, 2, 3 and 4). For each trial, the participant is required to identify the sorting rule which is valid for that trial, that is, which of the three feature has to be considered as a criterion to matching the target card with the right stimulus card (see Fig. 1). Notice that both features and sorting rules refer to the same concept. However, the feature still codifies a property of the card, whilst the sorting rule refers to the particular feature which is valid for the current trial.

Each response in the WCST is followed by a feedback informing the participant if his/her response is correct or incorrect. After some fixed number of consecutive responses, the sorting rule is changed by the experimenter without warning, and participants are required to infer the new sorting rule. Clearly, the most adaptive response would be to explore the remaining possible rules. However, participants sometimes would persist responding according to the old rule and produce what is called a *perseverative response*.

## METHODS

### The model

The core idea behind our computational framework is to encode the concept of *belief* into a generative probabilistic model of the environment. Belief updating then corresponds to recursive Bayesian updating of the internal model based on current and past interactions between the agent and its environment. Optimal or sub-optimal actions are selected according to a well specified or a misspecified internal model and, in turn, cause perceptible changes in the environment.

We assume that the cognitive agent aims to infer the *true hidden state* of the environment by processing and integrating sensory information from the environment. Within the context of the WCST, the hidden environmental states might change as a function of both the structure of the task and the (often sub-optimal) behavioral dynamics, so the agent constantly needs to rely on environmental feedback and own actions to infer the current state. We assume that the agent maintains an internal probability distribution over the states at each individual trial of the WCST. The agent then updates this distribution upon making new observations. In particular, the hidden environmental states to be inferred are the three features, $s_t \in \{1, 2, 3\}$, which refer the three possible sorting rules in the task environment such that 1: color, 2: shape and 3: number of objects. The posterior probability of the states depends on an observation vector $x_t = (a_t, f_t)$, which consists of the pair of agent's response $a_t \in \{1, 2, 3, 4\}$, codifying the action of choosing deck 1, 2, 3 or 4, and received feedback $f_t \in \{0, 1\}$, referring to the fact that a given response results in a failure (0) or in a success (1), in a given trial $t = 0, \ldots, T$. The discrete response $a_t$ represents the stimulus card indicator being matched with a target card at trial $t$. We denote a sequence of observations as $x_{0:t} = (x_0, x_1, \ldots, x_t) = ((a_0, f_0), (a_1, f_1), (a_2, f_2), \ldots, (a_t, f_t))$ and set $x_0 = \varnothing$ in order to indicate that there are no observations at the onset of the task. Thus, trial-by-trial belief updating is recursively computed according to Bayes' rule:

$$p(s_t | x_{0:t}) = \frac{p(x_t | s_t, x_{0:t-1}) p(s_t | x_{0:t-1})}{p(x_t | x_{0:t-1})}. \tag{1}$$

Accordingly, the agent's posterior belief about the task-relevant features $s_t$ after observing a sequence of response-feedback pairs $x_{0:t}$ is proportional to the product of the likelihood of observing a particular response-feedback pair and the agent's prior belief about the task-relevant feature in the current trial. The likelihood of an observation is computed as follows:

$$p(x_t | s_t, x_{0:t-1}) = \frac{f_t p(a_t | s_t = i) + (1 - f_t)(1 - p(a_t | s_t = i))}{f_t \sum_j p(a_t | s_t = j) + (1 - f_t) \sum_j (1 - p(a_t | s_t = j))} \tag{2}$$

where $j = 1, 2, 3$ and $p(a_t | s_t = i)$ indicates the probability of a matching between the target and the stimulus card assumed that the current feature is $i$. Here, we assume the likelihood of a current observation to be independent from previous observations without loss of generality, that is:

$$p(x_t | s_t, x_{0:t-1}) = p(x_t | s_t).$$

The prior belief for a given trial $t$ is computed based on the posterior belief generated in the previous trial, $p(s_{t-1}|x_{0:t-1})$, and the agent's belief about the probability of transitions between the hidden states, $p(s_t|s_{t-1})$. The prior belief can also be considered as a predictive probability over the hidden states. The predictive distribution for an upcoming trial $t$ is computed according to the Chapman–Kolmogorov equation:

$$p(s_{t+1}=k|x_{0:t}) = \sum_{i=1}^{3} p(s_{t+1}=k|s_t=i,\Gamma(t))p(s_t=i|x_{0:t}) \tag{3}$$

where $\Gamma(t)$ represents a stability matrix describing transitions between the states (to be explained shortly). Thus, the agent combines information from the updated belief (posterior distribution) and the belief about the transition properties of the environmental states to predict the most probable future state. The predictive distribution represents the internal model of the cognitive agent according to which actions are generated.

The stability matrix $\Gamma(t)$ encodes the agent's belief about the probability of states being stable or likely to change in the next trial. In other words, the stability matrix reflects the cognitive agent's internal representation of the dynamic probabilistic model of the task environment. It is computed on each trial based on the response-feedback pair, $x_t$, and a matching signal, $m_t$, which are observed.

The matching signal $m_t$ is a vector informing the cognitive agent which features are currently relevant (meaningful), such that $m_t^{(i)}=1$ when a positive feedback is associated with a response implying feature $s_t=i$, and $m_t^{(i)}=0$ otherwise. Note, that the matching signal is not a free parameter of the model, but is completely determined by the task contingencies. The matching signal vector allows the agent to compute the *state activation level* $\omega_t^{(i)} \in [0,1]$ for the hidden state $s_t=i$, which provides an internal measure of the (accumulated) evidence for each hidden state at trial $t$. Thus, the activation levels of the hidden states are represented by a vector $\omega_t$. The stability matrix is a square and asymmetric matrix related to hidden state activation levels such that:

$$\Gamma(t) = \begin{bmatrix} \omega_t^{(1)} & \frac{1}{2}(1-\omega_t^{(1)}) & \frac{1}{2}(1-\omega_t^{(1)}) \\ \frac{1}{2}(1-\omega_t^{(2)}) & \omega_t^{(2)} & \frac{1}{2}(1-\omega_t^{(2)}) \\ \frac{1}{2}(1-\omega_t^{(3)}) & \frac{1}{2}(1-\omega_t^{(3)}) & \omega_t^{(3)} \end{bmatrix} \tag{4}$$

where the entries $\Gamma_{ii}(t)$ in the main diagonal represent the elements of the activation vector $\omega_t$, and the non-diagonal elements are computed so as to ensure that rows sum to 1. The state activation vector is computed in each trial as follows:

$$\begin{bmatrix} \omega_t^{(1)} \\ \omega_t^{(2)} \\ \omega_t^{(3)} \end{bmatrix} = f_t \omega_{t-1}^{\delta} \begin{bmatrix} m_t^{(1)} \\ m_t^{(2)} \\ m_t^{(3)} \end{bmatrix} + \lambda \left[ (1-f_t)\omega_{t-1}^{\delta} \begin{bmatrix} 1-m_t^{(1)} \\ 1-m_t^{(2)} \\ 1-m_t^{(3)} \end{bmatrix} \right] \begin{bmatrix} \omega_{t-1}^{(1)} \\ \omega_{t-1}^{(2)} \\ \omega_{t-1}^{(3)} \end{bmatrix}. \tag{5}$$

This equation reflects the idea that state activations are simultaneously affected by the observed feedback, $f_t$, and the matching signal vector, $m_t$. However, the matching signal vector conveys different information based on the current feedback. Matching a target card

with a stimulus card makes a feature (or a subset of features) informative for a specific state. The vector $m_t$ contributes to increase the activation level of a state if the feature is informative for that state when a positive feedback is received, as well as to decrease the activation level when a negative feedback is received.

The parameter $\lambda \in [0, 1]$ modulates the efficiency to disengage attention to a given state-activation configuration when a negative feedback is processed. We therefore term this parameter *flexibility*. We also assume that information from the matching signal vector can degrade by slowing down the rate of evidence accumulation for the hidden states. This means that the matching signal vector can be re-scaled based on the current state activation level. The parameter $\delta \in [0, 1]$ is introduced to achieve this re-scaling. When $\delta = 0$, there is no re-scaling and updating of the state activation levels relies on the entire information conveyed by $m_t$. On the other extreme, when $\delta = 1$, several trials have to be accomplished before converging to a given configuration of the state activation levels. Equivalently, higher values of $\delta$ affect the entropy of the distribution over hidden states by decreasing the probability of sampling of the correct feature. We therefore refer to $\delta$ as *information loss*.

The free parameters $\lambda$ and $\delta$ are central to our computational model, since they regulate the rate at which the internal model converges to the true task environmental model. can be expressed in compact notation as follows:

$$\boldsymbol{\omega}_t = f_t \boldsymbol{\omega}_{t-1}^\delta m_t + \lambda \left[ (1 - f_t) \boldsymbol{\omega}_{t-1}^\delta (1 - m_t) \right] \boldsymbol{\omega}_{t-1}. \tag{6}$$

Note that the information loss parameter $\delta$ affects the amount of information that a cognitive agent acquires from environmental contingencies, irrespective of the type of feedback received. Global information loss thus affects the rate at which the divergence between the agent's internal model and the true model is minimized. Figure 2 illustrates these ideas.

The probabilistic representation of adaptive behaviour provided by our Bayesian agent model allows us to quantify latent cognitive dynamics by means of meaningful information-theoretic measures. Information theory has proven to be an effective and natural mathematical language to account for functional integration of structured cognitive processes and to relate them to brain activity (*Koechlin & Summerfield, 2007*; *Friston et al., 2017*; *Collell & Fauquet, 2015*; *Strange et al., 2005*; *Friston, 2003*). In particular, we are interested in three key measures, namely, *Bayesian surprise*, $\mathcal{B}_t$, *Shannon surprise*, $\mathcal{I}_t$, and *entropy*, $\mathcal{H}_t$. The subscript $t$ indicates that we can compute each quantity on a trial-by-trial basis. Each quantity is amenable to a specific interpretation in terms of separate neurocognitive processes. Bayesian surprise $\mathcal{B}_t$ quantifies the magnitude of the update from prior belief to posterior belief. Shannon surprise $\mathcal{I}_t$ quantifies the improbability of an observation given an agent's prior expectation. Finally, entropy $\mathcal{H}_t$ measures the degree of epistemic uncertainty regarding the true environmental states. Such measures are thought to account for the ability of the agent to manage uncertainty as emerging as a function of competing behavioral affordances (*Hirsh, Mar & Peterson, 2012*). We expect an adaptive system to attenuate uncertainty over environmental states (current features) by reducing the entropy of its internal probabilistic model.

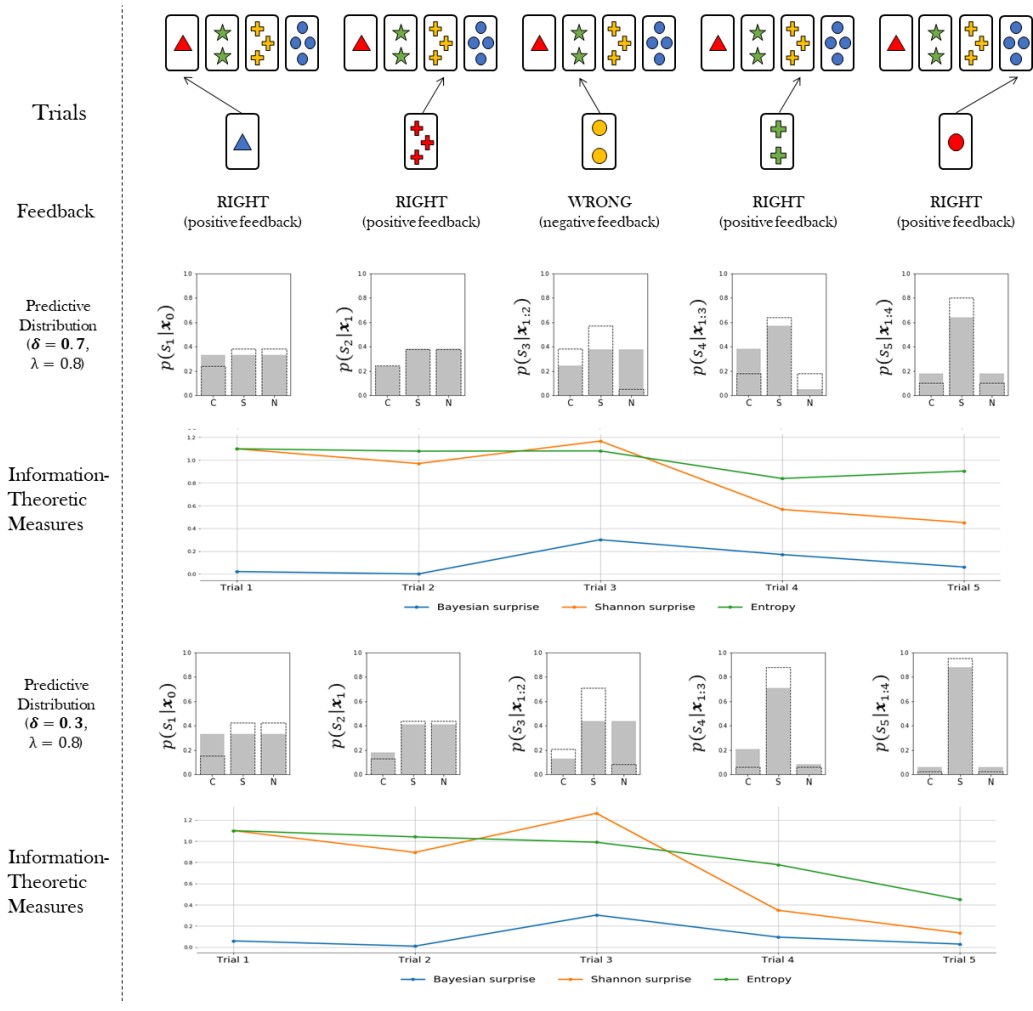

**Figure 2** The figure shows the rate of convergence of the predictive distributions to the true task environmental model. The predictive distributions at trial $t + 1$ depends on the sorting action $a_t$ (first row) and the received feedback $f_t$ (second row). Two examples of updating a predictive distribution are shown: one in which information loss is high ($\delta = 0.7$, third row), and one in which information loss is low ($\delta = 0.3$, fifth row). High information loss slows down the convergence of the internal model to the true environmental model. The gray bar plots represent the predictive probability distribution over the rules from which an action is sampled at each trial. Dotted bars represent the updated predictive distribution after the feedback observation. For each scenario, trial-by-trial information-theoretic measures are shown.

Bayesian surprise can be computed as the Kullback–Leibler ($\mathbb{KL}$) divergence between prior and posterior beliefs about the environmental states. Thus, Bayesian surprise accounts for the divergence between the predictive model for the current trial and the updated predictive model for the upcoming trial. It is computed as follows:

$$\mathcal{B}_t = \mathbb{KL}[p(s_{t+1}|x_{0:t})||p(s_t|x_{0:t-1})]$$
$$= \sum_{i=1}^{3}\left[p(s_{t+1}=i|x_{0:t})\log\left(\frac{p(s_{t+1}=i|x_{0:t})}{p(s_t=i|x_{0:t-1})}\right)\right] \tag{7}$$

The Shannon surprise of a current observation given a previous one is computed as the conditional information content of the observation:

$$\mathcal{I}_t = -\log p(x_t|x_{0:t-1})$$
$$= -\log \sum_{i=1}^{3} \left[ p(x_t|s_t = i)p(s_t = i|x_{0:t-1}) \right] \qquad (8)$$

Finally, the entropy is computed over the predictive distribution in order to account for the uncertainty in the internal model of the agent in trial $t$ as follows:

$$\mathcal{H}_t = \mathbb{E}\left[ -\log p(s_t|x_{0:t-1}) \right]$$
$$= -\sum_{i=1}^{3} p(s_t = i|x_{0:t-1}) \log p(s_t = i|x_{0:t-1}) \qquad (9)$$

Once the flexibility ($\lambda$) and information loss ($\delta$) parameters are estimated from data, the information-theoretic quantities can be easily computed and visualized for each trial of the WCST (see Fig. 2). This allows to rephrase standard neurocognitive constructs in terms of measurable information-theoretic quantities. Moreover, the dynamics of these quantities, as well as their interactions, can be used for formulating and testing hypotheses about the neurcognitive underpinnings of adaptive behavior in a principled way, as discussed later in the paper. A summary of all quantities relevant for our computational model is provided in Table 1.

## Simulations

In this section we evaluate the expressiveness of the model by assessing its ability to reproduce meaningful behavioral patterns as a function of its two free parameters. We study how the generative model behaves when performing the WCST in a 2-factorial simulated Monte Carlo design where flexibility ($\lambda$) and information loss ($\delta$) are systematically varied.

In this simulation, the Heaton version of the task (*Heaton, 1981*) is administered to the Bayesian cognitive agent. In this particular version, the sorting rule (true environmental state) changes after a fixed number of consecutive correct responses. In particular, when the agent correctly matches the target card in 10 consecutive trials, the sorting rule is automatically changed. The task ends after completing a maximum of 128 trials.

### Generative model

The cognitive agent's responses are generated at each time step (trial) by processing the experimental feedback. Its performance depends on the parameters governing the computation of the relevant quantities. The generative algorithm is outlined in Algorithm 1.

---

**Algorithm 1** Bayesian cognitive agent

1:    Set parameters $\boldsymbol{\theta} = (\lambda, \delta)$.

2:    Set initial activation levels $\boldsymbol{\omega}_0 = (0.5, 0.5, 0.5)$.

3:    Set initial observation $\boldsymbol{x}_0 = \varnothing$ and $p(s_1|\boldsymbol{x}_0) = p(s_1)$.

4:    **for** $t = 1, ..., T$ **do**

5:        Sample feature from prior/predictive internal model $s_t \sim p(s_t|\boldsymbol{x}_{0:t-1})$.

6:        Obtain a new observation $\boldsymbol{x}_t = (a_t, f_t)$.

7:        Compute state posterior $p(s_t|\boldsymbol{x}_{0:t})$.

8:        Compute new activation levels $\boldsymbol{\omega}_t$.

9:        Compute stability matrix $\boldsymbol{\Gamma}(t)$.

10:        Update prior/predictive internal model to $p(s_{t+1}|\boldsymbol{x}_{0:t})$.

11:    **end for**

---

***Simulation 1: clinical assessment of the Bayesian agent***

Ideally, the qualitative performance of the Bayesian cognitive agent will resemble human performance. To this aim, we adopt a metric which is usually employed in clinical assessment of test results in neurological and psychiatric patients (*Braff et al., 1991*; *Zakzanis, 1998*; *Bechara & Damasio, 2002*; *Landry & Al-Taie, 2016*). Thus, agent performance is codified according to a neuropsychological criterion (*Heaton, 1981*; *Flashman, Homer & Freides, 1991*) which allows to classify responses into several response types. These response types provide the scoring measures for the test.

Here, we are interested in: (1) non-perseverative errors (E); (2) perseverative errors (PE); (3) number of trials to complete the first category (TFC); and (4) number of failures to maintain set (FMS). Perseverative errors occur when the agent applies a sorting rule which was valid before the rule has been changed. Usually, detecting a perseveration error is far from trivial, since several response configurations could be observed when individuals are required to shift a sorting rule after completing a category (see *Flashman, Homer & Freides (1991)* for details). On the other hand, non-perseverative errors refer to all errors which do not fit the above description, or in other words, do not occur as a function of changing the sorting rule, such as casual errors.

The number of trials to complete the first category tells us how many trials the agent needs in order to achieve the first sorting principle, and can be seen as an index of conceptual ability (*Anderson, 2008*; *Singh, Aich & Bhattarai, 2017*). Finally, a failure to maintain a set occurs when the agent fails to match cards according to the sorting rule after it can be determined that the agent has acquired the rule. A given sorting rule is assumed to be acquired when the individual correctly sorts at least five cards in a row (*Heaton, 1981*; *Figueroa & Youmans, 2013*). Thus, a failure to maintain a set arises whenever a participant suddenly changes the sorting strategy in the absence of negative feedback. Failures to maintain a set are mostly attributed to distractibility. We compute this measure by counting the occurrences of first errors after the acquisition of a rule.

We run the generative model by varying flexibility across four levels, $\lambda \in \{0.3, 0.5, 0.7, 0.9\}$, and information loss across three levels, $\delta \in \{0.4, 0.7, 0.9\}$. We generate data from 150 synthetic cognitive agents per parameter combination and compute standard

**Table 1  Descriptive summary of all quantities involved in our model representation.**

| Expression | Name | Description |
|---|---|---|
| $s_t \in \{1, 2, 3\}$ | Sorting rule | Card feature relevant for the sorting criterion in trial $t$. |
| $a_t \in \{1, 2, 3, 4\}$ | Choice action | Action of choosing one of the four stimulus cards in trial $t$. |
| $f_t \in \{0, 1\}$ | Feedback | Indicates whether the action of matching a stimulus to a target card is correct or not in trial $t$. |
| $x_t = (a_t, f_t)$ | Observation | Pair of action and feedback which constitutes the agent's observation in trial $t$. |
| $\Gamma(t)$ | Stability matrix | Matrix encoding the agent's beliefs about state transitions from trial $t$ to the next trial $t+1$. |
| $\lambda \in [0, 1]$ | Flexibility | Parameter encoding the efficiency to disengage attention from a currently attended hidden state when signaled by the environment. |
| $\delta \in [0, 1]$ | Information loss | Parameter encoding how efficiently the agent's internal model converges to the true environmental model based on experience. |
| $m_t^{(i)} \in \{0, 1\}$ | Matching signal | Signal indicating whether feature $i$ is relevant in trial $t$ based on the feedback received. |
| $\omega_t^{(i)} \in [0, 1]$ | State activation level | Agent's internal measure of the accrued evidence for the hidden environmental state $i$ in trial $t$. |
| $\mathcal{B}_t \in \mathbb{R}^+$ | Bayesian surprise | Kullback–Leibler divergence between prior and posterior beliefs about hidden environmental states in trial $t$. |
| $\mathcal{I}_t \in \mathbb{R}^+$ | Shannon surprise | Information-theoretic surprise encoding the improbability or unexpectedness of an observation in trial $t$. |
| $\mathcal{H}_t \in \mathbb{R}^+$ | Entropy | Degree of epistemic uncertainty in the internal model of the environment in trial $t$. |

scoring measures for each of the agents simulated responses. Results from the simulation runs are depicted in Table 2 and a graphical representation is provided in Fig. 3.

The simulated performance of our Bayesian cognitive agents demonstrates that different parameter combinations capture different meaningful behavioral patterns. In other words, flexibility and information loss seem to interact in a theoretically meaningful way.

First, overall errors increase when flexibility ($\lambda$) decreases, which is reflected by the inverse relation between the number of casual, as well as perseverative, errors and the values of parameter $\lambda$. Moreover, this pattern is consistent across all the levels of parameter $\delta$. More precisely, information loss ($\delta$) seems to contribute to the characterization of the casual and the perseverative components of the error in a different way. Perseverative errors are likely to occur after a sorting rule has changed and reflect the inability of the agent to use feedback to disengage attention from the currently attended feature. They therefore result from local cognitive dynamics conditioned on a particular stage of the task (e.g., after completing a series of correct responses).

Second, information loss does not interact with flexibility when perseverative errors are considered. This is due to the fact that high information loss affects general performance by yielding a dysfunctional response strategy which increases the probability of making an error at any stage of the task. The lack of such interaction provides evidence that our

**Table 2  Mean clinical scoring measures as functions of flexibility ($\lambda$) and information loss ($\delta$). Cells show the average scores across simulated agents (standard deviation is shown in parenthesis).**

| Scoring measure | Info. Loss ($\delta$) | Flexibility ($\lambda$) | | | |
|---|---|---|---|---|---|
| | | $\lambda = 0.3$ | $\lambda = 0.5$ | $\lambda = 0.7$ | $\lambda = 0.9$ |
| Casual Errors (E) | $\delta = 0.4$ | 9.07 (2.68) | 7.95 (2.07) | 7.50 (2.13) | 6.85 (1.75) |
| | $\delta = 0.7$ | 10.84 (2.35) | 9.60 (2.2) | 8.25 (2.23) | 7.37 (1,74) |
| | $\delta = 0.9$ | 12.75 (2.96) | 11.25 (2.43) | 9.12 (2.09) | 7.79 (1.73) |
| Perseverative Errors (PE) | $\delta = 0.4$ | 20.81 (2.27) | 18.18 (1.88) | 14.99 (1.88) | 12.37 (1.12) |
| | $\delta = 0.7$ | 19.77 (2.55) | 17.65 (2.26) | 15.42 (1.94) | 12.39 (1.47) |
| | $\delta = 0.9$ | 18.56 (2.76) | 16.58 (2.53) | 14.49 (2.03) | 12.33 (1.44) |
| Trials to First Category (TFC) | $\delta = 0.4$ | 12.20 (1.46) | 11.91 (1.35) | 11.83 (1.24) | 11.67 (1.04) |
| | $\delta = 0.7$ | 13.82 (2.76) | 13.32 (2.52) | 12.97 (2.13) | 12.29 (1.53) |
| | $\delta = 0.9$ | 17.27 (4.21) | 16.63 (4.04) | 14.39 (3.58) | 12.91 (1.91) |
| Failures to Maintain Set (FMS) | $\delta = 0.4$ | 0.11 (0.31) | 0.09 (0.31) | 0.05 (0.32) | 0.02 (0.14) |
| | $\delta = 0.7$ | 1.65 (1.4) | 1.41 (1.3) | 0.84 (0.91) | 0.35 (0.69) |
| | $\delta = 0.9$ | 4.44 (1.96) | 3.88 (1.86) | 2.79 (1.56) | 1.54 (1.25) |

computational model can disentangle between error patterns due to perseveration and those due to general distractibility, according to neuropsychological scoring criteria.

However, in our framework, flexibility ($\lambda$) is allowed to yield more general and non-local cognitive dynamics as well. Indeed, $\lambda$ plays a role whenever belief updating is demanded as a function of negative feedback. An error classified as non-perseverative (e.g., casual error) by the scoring criteria might still be processed as a feedback-related evidence for belief updating. Consistently, the interaction between $\lambda$ and $\delta$ in accounting for causal errors shows that performance worsens when both flexibility and information loss become less optimal, and that such pattern becomes more pronounced for lower values of $\delta$.

On the other hand, a specific effect of information loss ($\delta$) can be observed for the scoring measures related to slow information processing and distractibility. The number of trials to achieve the first category reflects the efficiency of the agent in arriving at the first true environmental model. Flexibility does not contribute meaningfully to the accumulation of errors before completing the first category for some levels of information loss. This is reflected by the fact that the mean number of trials increases as a function of $\delta$, and do not change across levels of $\lambda$ for low and mid values of $\delta$. A similar pattern applies for failures to maintain a set. Both scoring measures index a deceleration of the process of evidence accumulation for a specific environmental configuration, although the latter is a more exhaustive measures of dysfunctional adaptation.

Therefore, an interaction between parameters can be observed when information loss is high. A slow internal model convergence process increases the amount of errors due to improper rule sampling from the internal environmental model. However, internal model convergence also plays a role when a new category has to be accomplished after completing an older one. On the one hand, compromised flexibility increases the amount of errors due to inefficient feedback processing. This leads to longer trial windows needed to achieve the first category. On the other hand, when information loss is high, belief updating upon

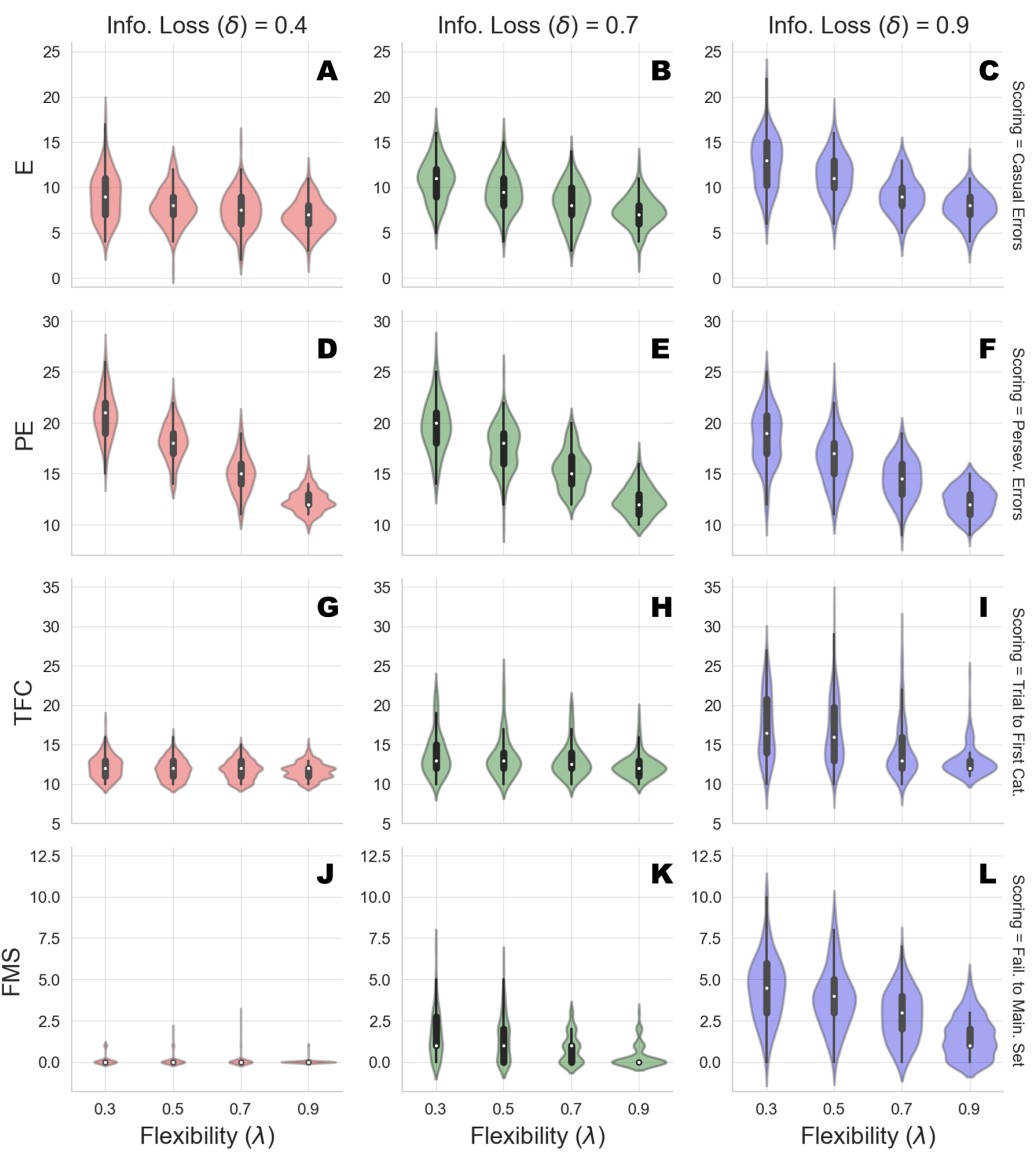

**Figure 3** **Clinical scoring measures as functions of flexibility (λ) and information loss (δ) - simulated scenarios.** The different cells show the violin plots for the estimated distribution densities of the scoring measures obtained from the group of synthetic individuals, for the levels of λ across different levels of δ. In particular, they show the distribution of non-perseverative errors (E: A–C), perseverative errors (PE: D–F), number of trials to complete the first category (TFC: G–I), number of failures to maintain set (FMS: J–L) obtained from 150 synthetic agent's response simulations for each cell of the factorial design.

negative feedback is compromised due to high internal model uncertainty. At this point, the probability to err due to distractibility increases, as accounted by the failures to maintain a set measures.

Finally, the joint effect of δ and λ for high levels of information loss suggests that the roles played by the two cognitive parameters in accounting for adaptive functioning can be entangled when neuropsychological scoring criteria are considered.

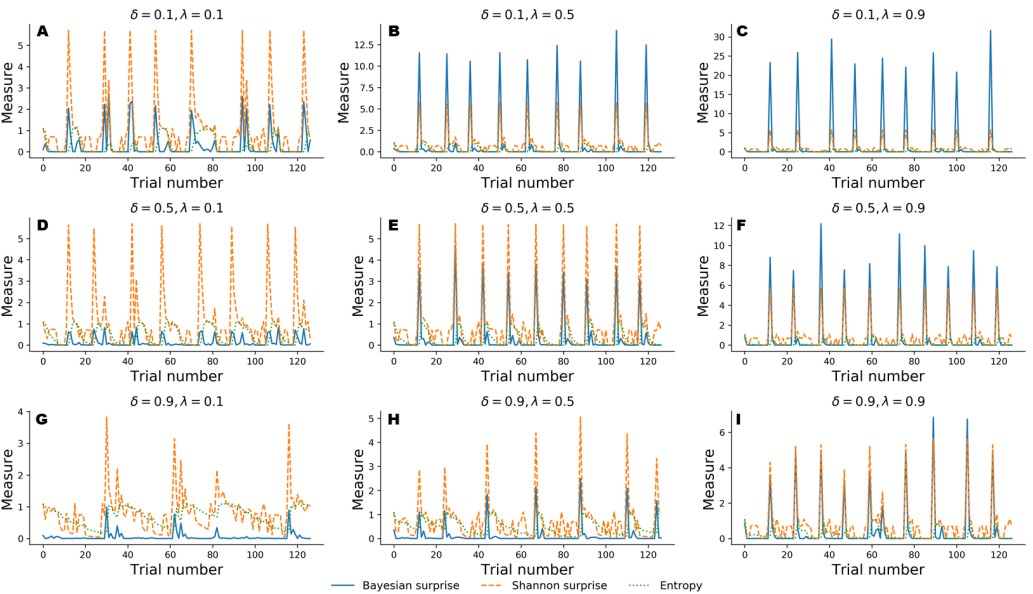

**Figure 4** **Information-theoretic measures varying as a function of flexibility λ and information loss δ across 128 trials of the WCST.** Trajectories depicted in A, D, and G show cognitive dynamics across the levels of information loss when flexibility is low. B, E, and H show the unfolding of information-theoretic quantities when flexibility is mildly impaired, whilst C, F, and I refer to an optimal flexibility value. Optimal belief updating and uncertainty reduction are achieved with low information loss and high flexibility (C).

### Simulation 2: Information-theoretic analysis of the Bayesian agent

In the following, we explore a different simulation scenario in which information-theoretic measures are derived to assess performance of the Bayesian cognitive agent. In particular, we explore the functional relationship between cognitive parameters and the dynamics of the recovered information-theoretic measures by simulating observed responses by varying flexibility across three levels, $\lambda \in \{0.1, 0.5, 0.9\}$, and information loss across three levels, $\delta \in \{0.1, 0.5, 0.9\}$.

For this simulation scenario, we make no prior assumptions about sub-types of error classification. Instead, we investigate the dynamic interplay between Bayesian surprise, $\mathcal{B}_t$, Shannon surprise, $\mathcal{I}_t$, and entropy, $\mathcal{H}_t$ over the entire course of 128 trials in the WCST.

Figure 4 depicts results from the nine simulation scenarios. Although an exhaustive discussion on cognitive dynamics should couple information-theoretic measures with patterns of correct and error responses, we focus solely on the information-theoretic time series for illustrative purposes. We refer to the 'Application' for a more detailed description of the relation between observed responses and estimated information-theoretic measures in the context of data from a real experiment.

Again, simulated performance of the Bayesian cognitive agent shows that different parameter combinations yield different patterns of cognitive dynamics. Observed spikes and their related magnitudes signal informative task events (e.g., unexpected negative feedback), as accounted by Shannon surprise, or belief updating, as accounted by Bayesian

surprise. Finally, entropy encodes the epistemic uncertainty about the environmental model on a trial-by-trial basis.

In general, low information loss ($\delta$) ensures optimal behavior by speeding up internal model convergence by decreasing the number of trials needed to minimize uncertainty about the environmental states. Low uncertainty reflects two main aspects of adaptive behavior. On the one hand, the probability that a response occurs due to sampling of improper rules decreases, allowing the agent to prevent random responses due to distractibility. On the other hand, model convergence entails a peaked Shannon surprise when a negative feedback occurs, due to the divergence between predicted and actual observations.

Flexibility ($\lambda$) plays a crucial role in integrating feedback information in order to enable belief updating. The first row depicted in Fig. 4 shows cognitive dynamics related to low information loss, across the levels of flexibility. As can be noticed, there is a positive relation between the magnitude of the Bayesian surprise and the level of flexibility, although unexpectedness yields approximately the same amount of signaling, as accounted by peaked Shannon surprise. From this perspective, surprise and belief updating can be considered functionally separable, where the first depends on the particular internal model probability configuration related to $\delta$, whilst the second depends on flexibility $\lambda$.

However, more interesting patterns can be observed when information loss increases. In particular, model convergence slows down and several trials are needed to minimize predictive model entropy. Casual errors might occur within trial windows characterized by high uncertainty, and interactions between entropy and Shannon surprise can be observes in such cases. In particular, Shannon surprise magnitude increases when model's entropy decreases, that is, during task phases in which the internal model has already converged. As a consequence, negative feedback could be classified as informative or uninformative, based on the uncertainty in the current internal model. This is reflected by the negative relation between entropy and Shannon surprise, as can be noticed by inspecting the graphs depicted in the third row of Fig. 4. Therefore, the magnitude of belief updating depends on the interplay between entropy and Shannon surprise, and can differ based on the values of the two measures in a particular task phase.

In sum, both simulation scenarios suggest that the simulated behavior of our generative model is in accord with theoretical expectations. Moreover, the flexibility and information loss parameters can account for a wide range of observed response patterns and inferred dynamics of information processing.

## Parameter estimation

In this section, we discuss the computational framework for estimating the parameters of our model from observed behavioral data. Parameter estimation is essential to inferring the cognitive dynamics underlying observed behavior in real-world applications of the model. This section is slightly more technical and can be skipped without significantly affecting the flow of the text.

### Computational framework

Rendering our cognitive model suitable for application in real-world contexts also entails accounting for uncertainty about parameter estimates. Indeed, uncertainty quantification turns out to be a fundamental and challenging goal when first-level quantities, that is, cognitive parameter estimates, are used to recover (second-level) information-theoretic measures of cognitive dynamics. The main difficulties arise when model complexity makes estimation and uncertainty quantification intractable at both analytical and numerical levels. For instance, in our case, probability distributions for the hidden model are generated at each trial, and the mapping between hidden states and responses changes depending on the structure of the task environment.

Identifying such a dynamic mapping is relatively easy from a generative perspective, but it becomes challenging, and almost impossible, when inverse modeling is required. Generally, this problem arises when the likelihood function relating model parameters to the data is not available in closed-form or too complex to be practically evaluated (*Sisson & Fan, 2011*). To overcome these limitations, we apply the first version of the recently developed *BayesFlow* method (see *Radev et al., 2020* for mathematical details). At a high-level, *BayesFlow* is a simulation-based method that estimates parameters and quantifies estimation uncertainty in a unified Bayesian probabilistic framework when inverting the generative model is intractable. The method is based on recent advances in deep generative modeling and makes no assumptions about the shape of the true parameter posteriors. Thus, our ultimate goal becomes to approximate and analyze the joint posterior distribution over the model parameters. The parameter posterior is given via an application of Bayes' rule:

$$p(\boldsymbol{\theta}|x_{0:T}, m_{0:T}) = \frac{p(x_{0:T}, m_{0:T}|\boldsymbol{\theta})p(\boldsymbol{\theta})}{\int p(x_{0:T}, m_{0:T}|\boldsymbol{\theta})p(\boldsymbol{\theta})d\boldsymbol{\theta}} \tag{10}$$

where we set $\boldsymbol{\theta} = (\lambda, \delta)$ and stack all observations and matching signals into the vectors $x_{0:T} = (x_0, x_1, \ldots, x_T)$ and $m_{0:T} = (m_0, m_1, \ldots, m_T)$, respectively. The *BayesFlow* method uses simulations from the generative model to optimize a neural density estimator which learns a probabilistic mapping between raw data and parameters. It relies on the fact that data can easily be simulated by repeatedly running the generative model with different parameter configurations $\boldsymbol{\theta}$ sampled from the prior. During training, the neural network estimator iteratively minimizes the divergence between the true posterior and an approximate posterior. Once the network has been trained, we can efficiently obtain samples from the approximate joint posterior distribution of the cognitive parameters of interest, which can be further processed in order to extract meaningful summary statistics (e.g., posterior means, medians, modes, etc.). Importantly, we can apply the same pre-trained inference network to an arbitrary number of real or simulated data sets (i.e., the training effort *amortizes* over multiple evaluations of the network).

For our purposes of validation and application, we train the network for 50 epochs which amount to 50000 forward simulations. As a prior, we use a bivariate continuous uniform distribution $p(\boldsymbol{\theta}) \sim \mathcal{U}([0,0],[1,1])$. We then validate performance on a separate validation set of 1000 simulated data sets with known *ground-truth* parameter values. Training the

networks took less than a day on a single machine with an NVIDIA® GTX1060 graphics card (CUDA version 10.0) using TensorFlow (version 1.13.1) (*Abadi et al., 2016*). In contrast, obtaining full parameter posteriors from the entire validation set took approximately 1.78 s. In what follows, we describe and report all performance validation metrics.

### Performance metrics and validation results

To assess the accuracy of point estimates, we compute the root mean squared error (RMSE) and the coefficient of determination ($R^2$) between posterior means and true parameter values. To assess the quality of the approximate posteriors, we compute a calibration error (*Radev et al., 2020*) of the empirical coverage of each marginal posterior Finally, we implement simulation-based calibration (SBC, *Talts et al., 2018*) for visually detecting systematic biases in the approximate posteriors.

*Point Estimates*. Point estimates obtained by posterior means as well as corresponding RMSE and $R^2$ metrics are depicted in Figs. 5A–5B. Note, that point estimates do not have any special status in Bayesian inference, as they could be misleading depending on the shape of the posteriors. However, they are simple to interpret and useful for ease-of-comparison. We observe that pointwise recovery of $\lambda$ is better than that of $\delta$. This is mainly due to suboptimal pointwise recovery in the lower $(0, 0.1)$ range of $\delta$. This pattern is evident in Figs. 5A–5B and is due to the fact that $\delta$ values in this range produce almost indistinguishable data patterns. Bootstrap estimates yielded an average RMSE of 0.155 ($SD = 0.004$) and an average $R^2$ of 0.708 ($SD = 0.015$) for the $\delta$ parameter. An average RMSE of 0.094 ($SD = 0.002$) and an average $R^2$ of 0.895 ($SD = 0.007$) were obtained for the $\lambda$ parameter. These results suggest good global pointwise recovery but also warrant the inspection of full posteriors, especially in the low ranges of $\delta$.

*Full Posteriors*. Average bootstrap calibration error was 0.011 ($SD = 0.005$) for the marginal posterior of $\delta$ and 0.014 ($SD = 0.007$) for the marginal posterior of $\lambda$. Calibration error is perhaps the most important metric here, as it measures potential under- or overconfidence across all confidence intervals of the approximate posterior (i.e., an $\alpha$-confidence interval should contain the true posterior with a probability of $\alpha$, for all $\alpha \in (0, 1)$). Thus, low calibration error indicates a faithful uncertainty representation of the approximate posteriors. Additionally, SBC-histograms are depicted in Figs. 5C–5D. As shown by *Talts et al. (2018)*, deviations from the uniformity of the rank statistic (also know as a PIT histogram) indicate systematic biases in the posterior estimates. A visual inspection of the histograms reveals that the posterior means slightly overestimate the true values of $\delta$. This corroborates the pattern seen in Figs. 5A–5B for the lower range of $\delta$.

Finally, Figs. 5E–5H depicts the full marginal posteriors on two example validation sets. Even on these two data sets, we observe strikingly different posterior shapes. The marginal posterior of $\delta$ obtained from the first data set is slightly left-skewed and has its density concentrated over the $(0.8, 1.0)$ range. On the other hand, the marginal posterior of $\delta$ from the second data set is noticeably right-skewed and peaked across the lower range of the parameter. The marginal posteriors of $\lambda$ appear more symmetric and warrant the use of the posterior mean as a useful summary of the distribution. These two examples underline the importance of investigating full posterior distributions as a means to encode

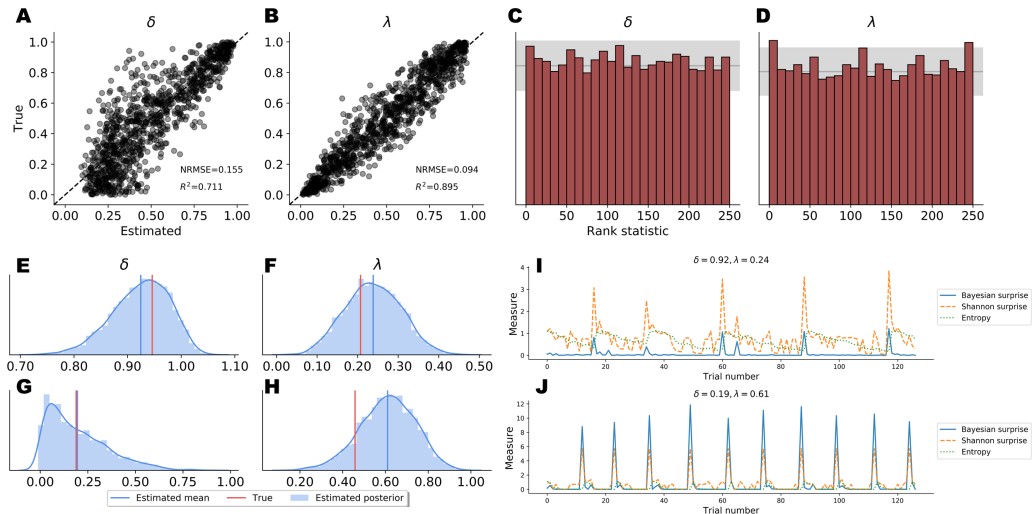

**Figure 5** Parameter recovery results on validation data; (A and B) posterior means vs. true parameter values; (C and D) histograms of the rank statistic used for simulation-based calibration; (E–H) example full posteriors for two validation data sets; (I and J) example information-theoretic dynamics recovered from the parameter posteriors.

epistemic uncertainty about parameter values. Moreover, they demonstrate the advantage of imposing no distributional assumptions on the resulting posteriors, as their form and sharpness can vary widely depending on the concrete data set.

# APPLICATION

In this section we fit the Bayesian cognitive model to real clinical data. The aim of this application is to evaluate the ability of our computational framework to account for dysfunctional cognitive dynamics of information processing in substance dependent individuals (SDI) as compared to healthy controls.

## Rationale

The advantage of modeling cognitive dynamics in individuals from a clinical population is that model predictions can be examined in light of available evidence about individual performance. For instance, SDIs are known to demonstrate inefficient conceptualization of the task and dysfunctional, error-prone response strategies. This has been attributed to defective error monitoring and behavior modulation systems, which depend on cingulate and frontal brain regions functionality (*Kübler, Murphy & Garavan, 2005*; *Willuhn, Sun & Steiner, 2003*). On the other hand, the WCST should be a rather easy and straightforward task for healthy participants to obtain excellent performance. Therefore, we expect our model to consistently capture such characteristics. To test these expectations, we estimate the two relevant parameters $\lambda$ and $\delta$ from both clinical patients and healthy controls from an already published dataset (*Bechara & Damasio, 2002*).

### The data

The dataset used in this application consists of responses collected by administering the standard Heaton version of the WCST (*Heaton, 1981*) to healthy participants and SDIs. In this version of the task, the sorting rule changes when a participant collects a series of 10 consecutive correct responses, and the task ends when this happens for 6 times. Participants in the study consisted of 39 SDIs and 49 healthy individuals. All participants were adults (>18 years old) and gave their informed consent for inclusion which was approved by the appropriate human subject committee at the University of Iowa. SDIs were diagnosed as substance dependent based on the Structured Clinical Interview for DSM-IV criteria (*First, 1997*).

### Model fitting

We fit the Bayesian cognitive agent separately to data from each participant in order to obtain individual-level posterior distributions. We apply the same *BayesFlow* network trained for the previous simulation studies, so obtaining posterior samples for each participant is almost instant (due to amortized inference).

### Results

The means of the joint posterior distributions are depicted for each individual in Fig. 6, and provide a complete overview of the heterogeneity in cognitive sub-components at both individual and group levels (individual-level full joint posterior distributions can be found in the Appendix A1).

The estimates reveal a rather interesting pattern across both healthy and SDI participants. In particular, in both clinical and control groups, individuals with a poor flexibility (e.g., low values of $\lambda$) can be detected. However, the group parameter space appears to be partitioned into two main clusters consisting of individuals with high and low flexibility, respectively. As can be noticed, the majority of SDIs belongs to the latter cluster, which suggests that the model is able to capture error-related defective behavior in the clinical population and attribute it specifically to the flexibility parameter. On the other hand, individual performance seems hardly separable along the information loss parameter dimension.

As a further validation, we compare the classification performance of two logistic regression models. The first uses the estimated parameter means as inputs and the participants' binary group assignment (patient vs. control) as an outcome. The second uses the four standard clinical measures (non-perseverative errors (E), perseverative errors (PE), number of trials to complete the first category (TFC), number of failures to maintain set (FMS) computed from the sample as inputs and the same outcome. Since we are interested solely in classification performance and want to mitigate potential overfitting due to small sample size, we compute leave-one-out cross-validated (LOO-CV) performance for both models. Interestingly, both logistic regression models achieve the same accuracy of 0.70, with a sensitivity of 0.71 and specificity of 0.70. Thus, it appears that our model is able to differentiate between SDIs and healthy individuals as good as the standard clinical measures.
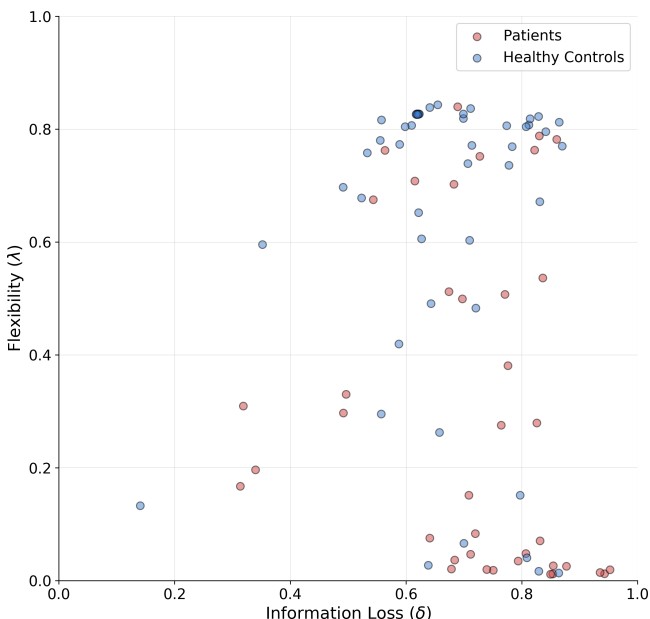

**Figure 6** **Joint posterior mean coordinates of the cognitive parameters, flexibility (λ) and information loss (δ), estimated for each individual.** We observe a great heterogeneity in the distribution of posterior means, most pronouncedly for the flexibility parameter. However, a moderate between-subject variability in information loss can still be observed in both groups.

However, as pointed out in the previous sections, estimated parameters serve merely as a basis to reconstruct cognitive dynamics by means of the trial-by-trial unfolding of information-theoretic measures. Moreover, cognitive dynamics can only be analysed and interpreted by relying on the joint contribution of both estimated parameters and individual-specific observed response patterns.

To further clarify this concept, we investigate the reconstructed time series of information-theoretic quantities based on the response patterns of two exemplary individuals (Fig. 7). In particular, Fig. 7A depicts the behavioral outcomes of a SDI with sub-optimal performance where the information-theoretic trajectories are reconstructed by taking the corresponding posterior means ($[\bar{\lambda} = 0.07, \bar{\delta} = 0.82]$), thus representing compromised flexibility and high information loss. Differently, Fig. 7B shows the information-theoretic path related to response dynamics of an optimal control participant, according to the parameter set $[\bar{\lambda} = 0.60, \bar{\delta} = 0.35]$, representing relatively high flexibility, and low information loss. Note, that in both cases, the reconstructed information-theoretic measures are based on the estimated posterior means for ease of comparison (see Appendix A1 for the full joint posterior densities of the two exemplary individuals and the rest of the sample).

Results in Fig. 7A account for a typical sub-optimal behavior observed in the SDI group, where several errors are produced in different phases of the task. The error patterns produced by such an individual might be induced by a non-trivial interaction between cognitive sub-components. Lower values of flexibility imply that errors are likely to be

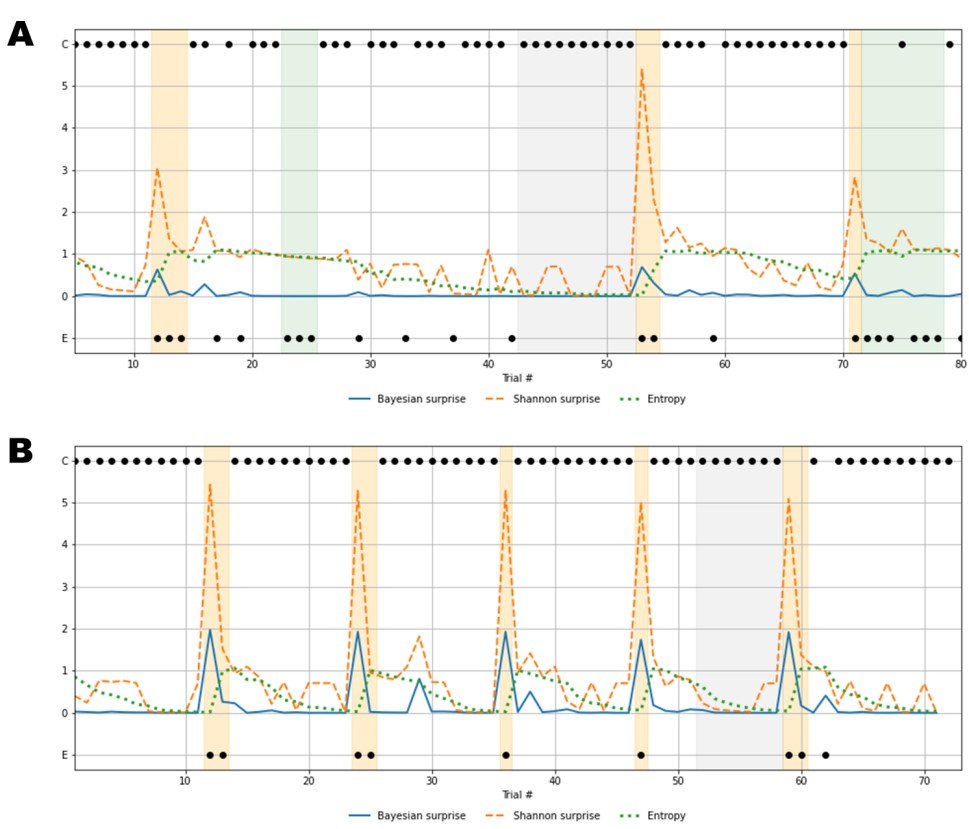

**Figure 7** **Recovered cognitive dynamics of two exemplary individuals.** (A) Trial-by-trial information-theoretic measures of a SDI characterized by very low flexibility and very high information loss; (B) trial-by-trial information-theoretic measures of a healthy individual characterized by relatively high flexibility and low information loss. Labels C and E in the y-axis indicate correct and error responses.

produced by generating responses from an internal environmental model which is no longer valid. In other words, the agent is unable to rely on local feedback-related information in order to update beliefs about hidden states. On the other hand, higher values of information loss reflect a general inefficiency of belief updating processes due to slow convergence to the optimal probabilistic environmental model. From this perspective, Bayesian surprise $\mathcal{B}_t$ and Shannon surprise $\mathcal{I}_t$ might play different roles in regulating behavior based on different internal model probability configurations. In addition, errors might be processed differently based on the status of the internal environmental representation, as reflected by the entropy of the predictive model, $\mathcal{H}_t$. Thus, information-theoretic measures allow to describe cognitive dynamics on a trial-by-trial basis and, further, to disentangle the effect that different feedback-related information processing dynamics exert on adaptive behavior.

Processing unexpected observations is accounted by the quantification of surprise upon observing a response-feedback pair which is inconsistent with the current internal model of the task environment. Negative feedback is maximally informative when errors occur after the internal model has converged to the true task model (grey area, Fig. 7A), or

the entropy approaches zero (grey line, Fig. 7A). The Shannon surprise (orange line) is maximal when errors occur within trial windows in which the agent's uncertainty about environmental states is minimal (orange areas, Fig. 7A). However, internal model updates following an informative feedback are not optimally performed, which is reflected by very small Bayesian surprise (blue line, Fig. 7A). This can be attributed to impaired flexibility and reflects the fact that after internal model convergence, informative feedback is not processed adequately and the internal model becomes impervious to change.

Conversely, errors occurring when the agent is uncertain about the true environmental state carry no useful information for belief updating, since the system fails to conceive such errors as unexpected and informative. The information loss parameter plays a crucial role in characterizing this cognitive behavior. The slow convergence to the true environmental model, accompanied by the slow reduction of entropy in the predictive model, leads to a large number of trials required to achieve a good representation of the current task environment (white areas, Fig. 7A). Errors occurring within trial windows with large predictive model entropy (green area, Fig. 7A) do not affect subsequent behavior, and feedback is maximally uninformative.

Rather different cognitive dynamics can be observed in Fig. 7B, accounting for a typical optimal behavior where the errors produced fall within the trial windows which follow a rule completion (e.g., when the individual completes a sequence of 10 consecutive correct responses), and, thus, the environmental model becomes obsolete. However, the high flexibility, $\lambda$, allows to rely on local feedback-related information to suddenly update beliefs about the hidden states, that is, the most appropriate sorting rule. In this case, negative feedback become maximally informative after model convergence (grey area, Fig. 7B) and the process of entropy reduction (green line, Fig. 7B) is faster (e.g., less trials are needed) compared to the sub-optimal behavior scenario. Since uncertainty about the environmental states decreases faster, the Shannon surprise is always highly peaked when errors occur (orange line, Fig. 7B), thus ensuring an efficient employment of the local feedback-related information. Accordingly, higher values of Bayesian surprise are observed (blue line, Fig. 7B), revealing optimal internal model updating.

In general, the role that predictive (internal) model uncertainty plays in characterizing the way the agent processes feedback allows to disentangle sub-types of errors based on the information they convey for subsequent belief updating. From this perspective, error classification is entirely dependent on the status of the internal environmental model across task phases. Identifying such a dynamic latent process is therefore fundamental, since the error codification criterion evolves with respect to the internal information processing dynamics. Otherwise, the problem of inferring which errors are due to perseverance in maintaining an older (converged) internal model and which due to uncertainty about the true environmental state becomes intractable, or even impossible.

## DISCUSSION

Investigating information processing related to changing environmental contingencies is fundamental to understanding adaptive behavior. For this purpose, cognitive scientists

mostly rely on controlled settings in which individuals are asked to accomplish (possibly) highly demanding tasks whose demands are assumed to resemble those of natural environments. Even in the most trivial cases, such as the WCST, optimal performance requires integrated and distributed neurocognitive processes. Moreover, these processes are unlikely to be isolated by simple scoring or aggregate performance measures.

In the current work, we developed and validated a new computational Bayesian model which maps distinct cognitive processes into separable information-theoretic constructs underlying observed adaptive behavior. We argue that these constructs could help describe and investigate the neurocognitive processes underlying adaptive behavior in a principled way.

Furthermore, we couple our computational model with a novel neural density estimation method for simulation-based Bayesian inference (*Radev et al., 2020*). Accordingly, we can quantify the entire information contained in the data about the assumed cognitive parameters via a full joint posterior over plausible parameter values. Based on the joint posterior, a representative summary statistic can be computed to simulate the most plausible unfolding of information-theoretic quantities on a trial-by-trial basis.

Several computational models have been proposed to describe and explain performance in the WCST, ranging from behavioral (*Bishara et al., 2010*; *Gläscher, Adolphs & Tranel, 2019*; *Steinke et al., 2020*) to neural network models (*Dehaene & Changeux, 1991*; *Amos, 2000*; *Levine, Parks & Prueitt, 1993*; *Monchi, Taylor & Dagher, 2000*). These models aim to provide psychologically interpretable parameters or biologically inspired network structures, respectively, accounting for specific qualitative patterns of observed data. Behavioral models, in particular, abstract the main cognitive features underlying individual performance in the WCST according to different theoretical frameworks (e.g., attentional updating (*Bishara et al., 2010*)), or reinforcement learning (*Steinke et al., 2020*) and disentangle psychological sub-processes explaining observed task performance. However, the main advantage of our Bayesian model is that it provides both a cognitive and a measurement model which coexist within the overarching theoretical framework of Bayesian brain theories. More precisely, the presented model is specifically designed to capture trial-by-trial fluctuations in information processing as described by second-order information-theoretic quantities. The latter can be seen as a multivariate quantitative account of the interaction between the agent and its environment. Moreover, it is worth noting that such a model representation might not be applicable outside a Bayesian theoretical framework.

Even though our computational model is not a neural model, it might provide a suitable description of cognitive dynamics at a representational and/or a computational level (*Marr, 1982*). This description can then be related to neural functioning underlying adaptive behavioral. Indeed, there is some evidence to suggest that neural processes related to belief maintenance/updating and unexpectedness are crucial for performance in the WCST. In particular, brain circuits associated with cognitive control and belief formation, such as the parietal cortex and prefrontal regions, seem to share a functional basis with neural substrates involved in adaptive tasks (*Nour et al., 2018*). Prefrontal regions appear to mediate the relation between feedback and belief updating (*Lie et al., 2006*) and efficient

functioning in such brain structures seems to be heavily dependent on dopaminergic neuromodulation (*Ott & Nieder, 2019*). Moreover, the dopaminergic system plays a role in the processing of salient and unexpected environmental stimuli, in learning based on error-related information, and in evaluating candidate actions (*Nour et al., 2018*; *Daw et al., 2011*; *Gershman, 2018*). Accordingly, dopaminergic system functioning has been put in relation with performance in the WCST (*Hsieh et al., 2010*; *Rybakowski et al., 2005*) and shown to be critical for the main executive components involved in the task, that is, cognitive flexibility and set-shifting (*Bestmann et al., 2014*; *Stelzel et al., 2010*). Further, neural activity in the anterior cingulate cortex (ACC) is increased when a negative feedback occurs in the context of the WCST (*Lie et al., 2006*). This finding corroborates the view that the ACC is part of an error-detection network which allocates attentional resources to prevent future errors. The ACC might play a crucial role in adaptive functioning by encoding error-related or, more generally, feedback-related information. Thus, it could facilitate the updating of internal environmental models (*Rushworth & Behrens, 2008*).

The neurobiological evidence suggests that brain networks involved in the WCST might endow adaptive behavior by accounting for maintaining/updating of an internal model of the environment and efficient processing of unexpected information. Is it noteworthy, that these processing aspects are incorporated into our computational framework. At this point, we briefly outline the empirical and theoretical potentials of the proposed computational framework for investigating adaptive functioning and discuss future research vistas.

*Model-Based Neuroscience.* Recent studies have pointed out the advantage of simultaneously modeling and analyzing neural and behavioral data within a joint modeling framework. In this way, the latter can be used to provide information for the former, as well as the other way around (*Turner et al., 2017*; *Turner et al., 2013*; *Forstmann et al., 2011*). This involves the development of joint models which encode assumptions about the probabilistic relationships between neural and cognitive parameters.

Within our framework, the reconstruction of information-theoretic discrete time series yields a quantitative account of the agent's internal processing of environmental information. Event-related cognitive measures of belief updating, epistemic uncertainty and surprise can be put in relation with neural measurements by explicitly providing a formal account of the statistical dependencies between neural and cognitive (information-theoretic) quantities. In this way, latent cognitive dynamics can be directly related to neural event-related measures (e.g., fMRI, EEG). Applications in which information-theoretic measures are treated as dependent variables in standard statistical analysis are also possible.

*Neurological Assessment.* Although neuroscientists have considered performance in the WCST as a proxy for measuring high-level cognitive processes, the usual approach to the analysis of human adaptive behavior consists in summarizing response patterns by simple heuristic scoring measures (e.g., occurrences of correct responses and sub-types of errors produced) and classification rules (*Flashman, Homer & Freides, 1991*). However, the theoretical utility of such a summary approach remains questionable. Indeed, adaptive behavior appears to depend on a complex and intricate interplay between multiple network structures (*Barcelo et al., 2006*; *Monchi et al., 2001*; *Lie et al., 2006*; *Barceló & Rubia, 1998*; *Buchsbaum et al., 2005*). This posits a great challenge for disentangling high-level cognitive

constructs at a model level and further investigating their relationship with neurobiological substrates. It appears that standard scoring measures might not be able to fulfil these tasks. Moreover, there is a pronounced lack of anatomical specificity in previous research concerning the neural and functional substrates of the WCST (*Nyhus & Barceló, 2009*).

Thus, there is a need for more sophisticated modeling approaches. For instance, disentangling errors due to perseverative processing of previously relevant environmental models from those due to uncertainty about task environmental states, is important and nontrivial. Sparse and distributed error patterns might depend on several internal model probability configurations. Such internal models are latent, and can only be uncovered through cognitive modeling. Therefore, information-based criteria to response (error) classification can enrich clinical evaluation beyond heuristically motivated criteria.

*Generalizability*. Another important advantage of the proposed computational framework is that it is not solely confined to the WCST. In fact, one can argue that the seventy-year old WCST does not provide the only or even the most suitable setting for extracting information about cognitive dynamics from general populations or maladaptive behavior in clinical populations. One can envision tasks which embody probabilistic (uncertain) or even chaotic environments (for instance with partially observable or unreliable feedback or partially observable states) and demand integrating information from different modalities (*O'Reilly et al., 2013*; *Nour et al., 2018*). These settings might prove more suitable for investigating changes in uncertainty-related processing or cross-modal integration than deterministic and fully observable WCST-like settings.

Despite these advantages, our proposed computational framework has certain limitations. A first limitation might concern the fact that the new Bayesian cognitive model accounts for the main dynamics in adaptive tasks by relying on only two parameters. Although such a parsimonious proposal suffices to disentangle latent data-generating processes, a more exhaustive formal description of cognitive sub-components might be envisioned. However, parameter estimation can become challenging in such a scenario, especially when one-dimensional response data is used as a basis for parameter recovery. Second, the information loss parameter appears to be more challenging to estimate than the flexibility parameter in some datasets. There are at least two possible remedies for this problem. On the one hand, global estimation of information loss might be hampered due to the model's current functional (algorithmic) formulation and can therefore be optimized via an alternative formulation/parameterization. On the other hand, it might be the case that the data obtainable in the simple WCST environment is not particularly informative about this parameter and, in general, not suitable for modeling more complex and non-linear cognitive dynamics in general. Future works should therefore focus on designing and exploring more data-rich controlled environments which can provide a better starting point for investigating complex latent cognitive dynamics in a principled way. Additionally, the information loss parameter seems to be less effective in differentiating between substance abusers and healthy controls in the particular sample used in this work. Thus, further model-based analyses on individuals from different clinical populations are needed to fully understand the potential of our 2-parameter model as a clinical neuropsychological tool. Finally, in this work, we did not perform formal model comparison, as this would

require an extensive consideration of various nested and non-nested model within the same theoretical framework and between different theoretical frameworks. We therefore leave this important endeavor for future research.

## CONCLUSIONS

In conclusion, the proposed model can be considered as the basis for a (bio)psychometric tool for measuring the dynamics of cognitive processes under changing environmental demands. Furthermore, it can be seen as a step towards a theory-based framework for investigating the relation between such cognitive measures and their neural underpinnings. Further investigations are needed to refine the proposed computational model and systematically explore the advantages of the Bayesian brain theoretical framework for empirical research on high-level cognition.

## ACKNOWLEDGEMENTS

We thank Karin Prillinger and Luca D'Alessandro for reading the manuscript and providing useful suggestions which significantly improved the original text.

### Funding

The authors received no funding for this work. Stefan T. Radev was supported by the Deutsche Forschungsgemeinschaft (DFG, German Research Foundation; grant number GRK 2277 "Statistical Modeling in Psychology").

### Grant Disclosures

The following grant information was disclosed by the authors:
Deutsche Forschungsgemeinschaft: GRK 2277.

### Competing Interests

The authors declare there are no competing interests.

### Author Contributions

- Marco D'Alessandro and Stefan Radev conceived and designed the experiments, performed the experiments, analyzed the data, prepared figures and/or tables, authored or reviewed drafts of the paper, and approved the final draft.
- Andreas Voss and Luigi Lombardi conceived and designed the experiments, authored or reviewed drafts of the paper, and approved the final draft.

### Data Availability

The codes for generating data and estimate parameters are available at GitHub: https://github.com/stefanradev93/DBN.

The codes of the generative model and parameter estimation are available in "WCST_INN_Final.ipynb". The data used for parameter estimation are in

"Data128.ipynb" and "MathingMat.ipynb". The codes for producing the information-theoretic measures from estimated parameters are shown in "ITmeasures.ipynb". The codes require the library Bayes Flow (https://github.com/stefanradev93/BayesFlow).

## Supplemental Information

Supplemental information for this article can be found online at http://dx.doi.org/10.7717/peerj.10316#supplemental-information.

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

# PeerJ

**Levine DS, Parks RW, Prueitt PS. 1993.** Methodological and theoretical issues in neural network models of frontal cognitive functions. *International Journal of Neuroscience* **72(3-4)**:209–233 DOI 10.3109/00207459309024110.

**Lie C-H, Specht K, Marshall JC, Fink GR. 2006.** Using fMRI to decompose the neural processes underlying the Wisconsin Card Sorting Test. *NeuroImage* **30(3)**:1038–1049 DOI 10.1016/j.neuroimage.2005.10.031.

**Marr D. 1982.** *Vision: a computational investigation into the human representation and processing of visual information*. San Francisco: W.H. Freeman.

**Monchi O, Petrides M, Petre V, Worsley K, Dagher A. 2001.** Wisconsin Card Sorting revisited: distinct neural circuits participating in different stages of the task identified by event-related functional magnetic resonance imaging. *Journal of Neuroscience* **21(19)**:7733–7741 DOI 10.1523/JNEUROSCI.21-19-07733.2001.

**Monchi O, Taylor JG, Dagher A. 2000.** A neural model of working memory processes in normal subjects, Parkinson's disease and schizophrenia for fMRI design and predictions. *Neural Networks* **13(8–9)**:953–973 DOI 10.1016/S0893-6080(00)00058-7.

**Nour MM, Dahoun T, Schwartenbeck P, Adams RA, FitzGerald TH, Coello C, Wall MB, Dolan RJ, Howes OD. 2018.** Dopaminergic basis for signaling belief updates, but not surprise, and the link to paranoia. *Proceedings of the National Academy of Sciences of the United States of America* **115(43)**:E10167–E10176 DOI 10.1073/pnas.1809298115.

**Nyhus E, Barceló F. 2009.** The Wisconsin Card Sorting Test and the cognitive assessment of prefrontal executive functions: a critical update. *Brain and Cognition* **71(3)**:437–451 DOI 10.1016/j.bandc.2009.03.005.

**Ott T, Nieder A. 2019.** Dopamine and cognitive control in prefrontal cortex. *Trends in Cognitive Sciences* **23(3)**:213–234.

**O'Reilly JX, Schüffelgen U, Cuell SF, Behrens TE, Mars RB, Rushworth MF. 2013.** Dissociable effects of surprise and model update in parietal and anterior cingulate cortex. *Proceedings of the National Academy of Sciences* **110(38)**:E3660–E3669 DOI 10.1073/pnas.1305373110.

**Petzschner FH, Glasauer S, Stephan KE. 2015.** A Bayesian perspective on magnitude estimation. *Trends in Cognitive Sciences* **19(5)**:285–293 DOI 10.1016/j.tics.2015.03.002.

**Radev ST, Mertens UK, Voss A, Ardizzone L, Kthe U. 2020.** BayesFlow: Learning complex stochastic models with invertible neural networks. ArXiv preprint. arXiv:2003.06281.

**Rushworth MF, Behrens TE. 2008.** Choice, uncertainty and value in prefrontal and cingulate cortex. *Nature Neuroscience* **11(4)**:389–397 DOI 10.1038/nn2066.

**Rybakowski J, Borkowska A, Czerski P, Kapelski P, Dmitrzak-Weglarz M, Hauser J. 2005.** An association study of dopamine receptors polymorphisms and the Wisconsin Card Sorting Test in schizophrenia. *Journal of Neural Transmission* **112(11)**:1575–1582 DOI 10.1007/s00702-005-0292-6.

**Sayood K. 2018.** Information theory and cognition: a review. *Entropy* **20(9)**:706 DOI 10.3390/e20090706.

**Schwartenbeck P, FitzGerald TH, Dolan R. 2016.** Neural signals encoding shifts in beliefs. *NeuroImage* **125**:578–586 DOI 10.1016/j.neuroimage.2015.10.067.

**Singh S, Aich TK, Bhattarai R. 2017.** Wisconsin Card Sorting Test performance impairment in schizophrenia: an Indian study report. *Indian journal of psychiatry* **59(1)**:88–93 DOI 10.4103/0019-5545.204440.

**Sisson SA, Fan Y. 2011.** *Likelihood-free MCMC.* Chapman & Hall/CRC, New York.[839].

**Steinke A, Lange F, Seer C, Hendel MK, Kopp B. 2020.** Computational modeling for neuropsychological assessment of bradyphrenia in Parkinsons disease. *Journal of Clinical Medicine* **9(4)**:1158 DOI 10.3390/jcm9041158.

**Stelzel C, Basten U, Montag C, Reuter M, Fiebach CJ. 2010.** Frontostriatal involvement in task switching depends on genetic differences in d2 receptor density. *Journal of Neuroscience* **30(42)**:14205–14212 DOI 10.1523/JNEUROSCI.1062-10.2010.

**Stephan KE, Baldeweg T, Friston KJ. 2006.** Synaptic plasticity and dysconnection in schizophrenia. *Biological Psychiatry* **59(10)**:929–939 DOI 10.1016/j.biopsych.2005.10.005.

**Strange BA, Duggins A, Penny W, Dolan RJ, Friston KJ. 2005.** Information theory, novelty and hippocampal responses: unpredicted or unpredictable? *Neural Networks* **18(3)**:225–230 DOI 10.1016/j.neunet.2004.12.004.

**Sun R. 2009.** Theoretical status of computational cognitive modeling. *Cognitive Systems Research* **10(2)**:124–140 DOI 10.1016/j.cogsys.2008.07.002.

**Talts S, Betancourt M, Simpson D, Vehtari A, Gelman A. 2018.** Validating Bayesian inference algorithms with simulation-based calibration. ArXiv preprint. arXiv:1804.06788.

**Turner BM, Forstmann BU, Love BC, Palmeri TJ, Van Maanen L. 2017.** Approaches to analysis in model-based cognitive neuroscience. *Journal of Mathematical Psychology* **76**:65–79 DOI 10.1016/j.jmp.2016.01.001.

**Turner BM, Forstmann BU, Wagenmakers E-J, Brown SD, Sederberg PB, Steyvers M. 2013.** A Bayesian framework for simultaneously modeling neural and behavioral data. *NeuroImage* **72**:193–206 DOI 10.1016/j.neuroimage.2013.01.048.

**Willuhn I, Sun W, Steiner H. 2003.** Topography of cocaine-induced gene regulation in the rat striatum: relationship to cortical inputs and role of behavioural context. *European Journal of Neuroscience* **17(5)**:1053–1066 DOI 10.1046/j.1460-9568.2003.02525.x.

**Zakzanis KK. 1998.** The subcortical dementia of Huntington's disease. *Journal of Clinical and Experimental Neuropsychology* **20(4)**:565–578 DOI 10.1076/jcen.20.4.565.1468.