# Peer review of "A Bayesian brain model of adaptive behavior: an application to the Wisconsin Card Sorting Task"

_PeerJ, doi:10.7717/peerj.10316_

## Round 0.1 · original submission · Minor Revisions

Two independent reviewers have now submitted their comments, and they are both quite positive towards the model proposed by the authors.

However, both Reviewers provide some important suggestions on how to improve your manuscript in order to make it suitable for publication.

In particular, I believe that a model comparison would strengthen your conclusions and that it would also be useful to assess the performance in healthy controls.

I also agree with the overall observation that the manuscript should be made more accessible to a wider audience.

·

Basic reporting

Please, provide a more detailed abstract that better clarify the methodology and the conclusions of the article, that in this version are only mentioned.
You do not mention anything about the Bayesian models used. Moreover, writing “theoretical implications … are finally discussed” means not reporting anything concerning the discussion of the paper.
The limits of the abstract are 500 words, and 3000 characters. Your abstract has 218 words and it is 1628 characters long, therefore, there is space for improvement.

The Winsconsin Card Sorting Test is a well-known neuropsychological test, currently used in neuropsychological settings as well as in experimental psychology. In particular, in neuropsychology, this test is used to test clinical aspects of perseverance (typical of frontal lesions), decision making, abstract reasoning and so on. It is inherently connected with executive functions.
For this reason, I think that the Authors should make an effort to be more understandable for the clinical and experimental neuropsychologist, with further examples concerning the Bayesian brain theories (e.g. they can report a classic example concerning sensory processing), and which (if any) benefits the neuropsychologist could take from using this perspective.

I would avoid terms like “dub”. If you “dub” something, it should have a real name. The coefficients dubbed as “flexibility” and “information loss” have a specific name?

Page 3, lines 10-20. In the general description of the Winsconsin card sorting test, you should report which is the version that you are using, if a validation and normative data eixst, how many cards there are, after how many consecutive responses the sorting rule changes, etc.

Please, when you refer to BayesFlow, add the version number or the date of the version. This is necessary because it is your open-source software which will be very likely updated in the next future.

In Figure 4 and 5, the labels are too small.

In the caption of Figure 3 please report what the violin plots and the boxplot are showing. Please, make more visible the boxplot because it is very hard to see (maybe you could increase the alpha level of the colour that is filling the violin plots, and make larger the boxplots?).

Page 12, lines 395-398. Why are you reporting only the graphics card of the computer? If you have used a CUDA-based computing approach you should explicitly state it.

Please, check the text for typos.

Experimental design

Page 4, lines 32-40. Please, make it clearer to what the hidden environmental states, the vectors of observation, and the feedback are.
More specifically, what does it mean s_t = 1, or 2, or 3? This is very confusing because all the three features (colour, shape and number of objects) can change among 4 different states. So I would expect something like s_t =(c_t, sh_t, n_t), where is the colour, is the shape, is the number of symbols.
You are probably referring to the fact that the sorting rule can be shape, colour or number, but this is not specified anywhere. Please be clearer and use a mathematical symbology that is related to the task. If this is the case, you should call s “sorting rule”, and clearly state what s_t = 1, or 2, or 3 mean.
Please, also specify that is the action of choosing desk 1, 2, 3 or 4, and that is the outcome failure (0) or success (1).

Page 3, line 30. Please, make it clearer that “the hidden environmental states might change at a non-constant rate” means that they change each action of the agent.

In (2) I suppose that the j in the summation should be within {1, 2, 3}. Please, if I am correct, add it.

Page 5, rows 76 – 77: what do you mean with resp. decrease and resp. negative? You never introduced the “resp.” abbreviation.

Page 8, simulation 1. Please, report central tendency indexes and dispersion parameters of your choice in a table to numerically describe the results obtained from the simulations shown in Figure 3.

I think that it might be interesting to see when E, PE, TCG and FMS in one condition are different from the other conditions. Would you mind to add statistical inference about that (better if done in the Bayesian framework)?

Also in simulation 2, a statistical comparison among the different results would be nice.

Page 14, Results. The results are very interesting, however, I am wondering why we do not have the results also from the Healthy population to make a comparison for the flexibility, information loss parameters, Bayesian surprise, Shannon surprise, and entropy.

Validity of the findings

no comment

·

Basic reporting

[Minor comments]
Overall, both in the figures and text the authors could write what the parameters stand for instead of just the parameter itself. This is performed for example in the legend of figure 6 (..flexibility(lambda) and information loss(delta)) but not elsewhere. This is important particularly for readers with (currently) low working memory, so they don’t have to be searching for the meaning of the parameters all the time. Alternatively (or in addition), the authors could include a table with the parameter, its name, and what it measures (+ the space of values that it can have).
- Figure 4: the green and blue lines are sometimes hard to distinguish, so consider having some of the lines dashed or dotted.
- Figure 6: as stated above, it would be nice to also have examples of figures from control participants.

Experimental design

No comment

Validity of the findings

No comment

Additional comments

Alessandro and colleagues develop a Bayesian model that can be applied to the Wisconsin Card Sorting Task and has the potential to be applicable to other tasks as well. This is a nicely-written, thorough paper, that gives a potentially useful new tool to analyze a very commonly used task. The main advantage of their model is that it is normatively based and uses behavioral data from all trials, rather than relying only on summary statistics. I have only some comments/suggestions:
- The authors fit their model to actual data of substance-dependent individuals (SDI). This allowed them to see that SDIs generally had low flexibility and high information loss (as per the parameters of their model). It would be useful to see how their model performs in healthy controls, and what the model parameters are in this group.
- To the point above, it would be interesting to see if the model could have a direct clinical utility in terms of distinguishing between SDIs and healthy controls. It is, however, a useful model to have even if it doesn’t distinguish between them.
- How does the model compare to existing models in the literature (e.g. a Reinforcement learning model)? However, it is perfectly ok for the authors to decide that model comparison is not an objective of this paper, in which case this could be stated directly (the discussion section suggests it but it would be good to have it explicitly written earlier in the paper too).

Overall this is a good paper, with a model that has the potential to be used broadly.

---

## Round 0.2 · Minor Revisions

I am glad to communicate that your manuscript is suitable for publication on PeerJ, pending on minor changes, as requested by the reviewers.

·

Basic reporting

This part of the article is improved.

Experimental design

This part of the article is improved.

Validity of the findings

This part of the article is improved.

Additional comments

Everything is fine by my side, please only check your references.
Probably you used an old bibtex file because there are a lot of missing references (e.g. page 2, lines 79-82).

·

Basic reporting

• There are a few question marks instead of possible citations (e.g. neurocognitive mechanisms, such as motor control (?), oculomotor dynamics70 (?), object recognition (?), attention (?) ). This is probably a reference manager/compilation issue, but should be corrected.

Experimental design

No comment

Validity of the findings

No comment

Additional comments

The manuscript has substantially improved. I only have a few minor suggestions/edits:
• In the example the authors give of a perceptual task, the authors could have chosen a more intuitive/more tangible example (e.g. judging if an item in the sky is a plane or a bird; the weight of a mug of coffee and the force needed to grab it without spilling; etc.). This is, of course, a personal preference, and the authors are free to just keep the example as is.
• There are a few question marks instead of possible citations (e.g. neurocognitive mechanisms, such as motor control (?), oculomotor dynamics70 (?), object recognition (?), attention (?) ). This is probably a reference manager/compilation issue but should be corrected.
• Just before Figure 1 – replace “noticed” by “notice” (“Noticed that both features and sorting rules refer to the same” – line 148 of the clean manuscript)

---

## Round 0.3 · accepted · Accept

I am pleased to announce your manuscript is now suitable for publication on PeerJ, since you successfully addressed the remaining minor issues raised by the reviewers.